# POISONING-BASED BACKDOOR ATTACKS FOR ARBITRARY TARGET LABEL WITH POSITIVE TRIGGERS

## ABSTRACT

Poisoning-based backdoor attacks expose vulnerabilities in the data preparation stage of deep neural network (DNN) training. The DNNs trained on the poisoned dataset will be embedded with a backdoor, making them behave well on clean data while outputting malicious predictions whenever a trigger is applied. To exploit the abundant information contained in the input data to output label mapping, our scheme utilizes the network trained from the clean dataset as a trigger generator to produce poisons that significantly raise the success rate of backdoor attacks versus conventional approaches. Specifically, we provide a new categorization of triggers inspired by the adversarial technique and develop a multi-label and multi-payload Poisoning-based backdoor attack with Positive Triggers (PPT), which effectively moves the input closer to the target label on benign classifiers. After the classifier is trained on the poisoned dataset, we can generate an input-label-aware trigger to make the infected classifier predict any given input to any target label with a high possibility. Under both dirty- and clean-label settings, we show empirically that the proposed poisoning-based backdoor attack achieves a high attack success rate without sacrificing accuracy across various datasets, including SVHN, CIFAR10, GTSRB, and Tiny ImageNet. Furthermore, the PPT attack can elude a variety of classical backdoor defenses, proving its effectiveness.

## 1 INTRODUCTION

Deep neural networks (DNNs) have exhibited groundbreaking successes in various applications such as computer vision He et al. (2016), natural language processing Devlin et al. (2018), etc. The vital capability of DNNs is partly attributed to a large amount of training data. However, some datasets are downloaded from the internet without reliability guarantee. This introduces a threat called poisoning-based backdoor attack Gu et al. (2017) that maliciously applies a specific trigger to a portion of the training data. Consequently, any model trained on this dataset will perform normally on clean data, but output unreasonable results when the trigger is present.

With the attacker's capacities gradually increasing, the backdoor attack can occur during the dataset preparation, training, and parameter or structure post-tuning phases. The more phases available to the attacker, the greater the probability of a successful attack. In this paper, we focus on planting backdoors in the image classification task through data poisoning, called poisoning-based backdoor, a less aggressive but more likely way to gain the user's trust than directly providing well-trained classifiers. The attacker can only manipulate the dataset but not interfere with the training process. Therefore, the pattern of the trigger is critical to the success of the attack. Since the introduction of backdoor attack, various patterns of triggers have sprung up in recent years. Patch-based triggers Gu et al. (2017); Liu et al. (2018b) design a specific black-and-white or color block for the target class and add it to the input at a specified location, which can be easily found out by human inspection. To increase the stealthiness of triggers, the blended method Chen et al. (2017); Liu et al. (2020) fuses inputs and triggers, and Doan et al. (2021a;b) utilizes the DNNs to generate invisible triggers with some limitations. In most cases, thanks to the strong capability of the DNNs, the backdoor attack makes the classifier overfit the link between the pre-defined or generated triggers supervised by several hyperparameters and target labels. These approaches ignore the mapping from the inputs to labels, which guarantees the accuracy of the clean dataset, and independently establish the trigger-label link within the classifier. This wastes resources because the input-label mapping instilled in a clean-data pre-trained network contains abundant prior information to help design triggers. We explore trigger

types based on the input-label link and divide them into three categories based on whether they can promote or restrain the input to be classified into the target class.

Previous backdoor attack research is mainly divided into two categories in terms of attack targets: one is *all2one*, where any input with a trigger is classified into one target label, and the other is *all2all*, where inputs of different classes are recognized as a prescribed target label. As stated in Marksman Doan et al. (2022), *all2all* is a special single-trigger single-payload attack, since it can only modify the inputs within one true label into one target label. A new backdoor attack that does not fall within these two categories is proposed by Marksman, called the multi-label and multi-payload backdoor attack. It can misclassify any input to any target label, which is nontrivial since it requires injecting a backdoor for each class into the model. As the number of classes increases, the accuracy and the attack success rate will decrease. Marksman Doan et al. (2022) uses the trigger function to generate the invisible input-label-aware triggers and build a solid link between the trigger function and the label by alternating optimization of the trigger function and classifier. However, it needs to control the whole training process of the classifier, which is impossible for the poisoning-based backdoor attack. So, there is still a gap for poisoning-based backdoor attacks that can arbitrarily select a target label that the classifier trained on the poisoned dataset will misclassify given any input.

In this paper, we extend the multi-label and multi-payload backdoor attack to the poisoning-based scenario where the stages available to the attacker are much fewer than Marksman. Based on the input-label links, we define triggers that can reduce classification loss to the target labels as positive ones and use them as the basis for our design. Different from the universal adversarial patch-based triggers Zhao et al. (2020) crafted by minimizing the loss for each class, which is human-perceptible and cannot break STRIP and spectral detection defenses, we generate the input-label-aware invisible triggers utilizing the adversarial targeted technique. The magnitude of the triggers is restricted by the $l_p$ norm to ensure stealthiness. Numerous experiments have shown that the PPT achieves superior accuracy and attack success rate compared to other poisoning-based methods in the multi-label and multi-payload backdoor attack field, under both dirty- and clean-label settings.

Our main contributions are summarized threefold:

1. We propose a new categorization of poisoning triggers, namely, the positive, neutral and negative triggers. These triggers can be leveraged to flexibly manipulate network classification results, increasing or decreasing the prediction scores of the target class.

2. Our method can achieve a multi-label and multi-payload backdoor attack with lower requirements than the existing method. Only by poisoning a portion of the clean dataset, PPT can cover both dirty- and clean-label attacks simultaneously.

3. We empirically prove the effectiveness of the PPT backdoor attack and the robustness against several popular defences. The classifier trained naturally on the poisoned dataset generated by PPT performs well on the clean dataset and has a high attack success rate of misclassifying the input into any arbitrary label when the trigger is present.

## 2 RELATED WORKS

### 2.1 BACKDOOR ATTACK

Backdoor attack, which exposes the vulnerability of DNNs during the training process, is an emerging and rapidly growing research area in recent years. In the image classification field, the classifier implanted with a backdoor performs well on clean dataset, but incorrectly classifies the input as the target class whenever the trigger is present. The trigger is the important key to activate the backdoor inside of the classifier, especially for poisoning-based backdoor attacks. There are a variety of criteria to categorize triggers Li et al. (2022), such as, trigger visibility, trigger selection, and trigger appearance. None of these categorization takes into account the relationship between triggers and input-label links that needs to be established to gain accuracy. To fill this gap, we divide triggers into three categories: positive, neutral, and negative triggers, according to the input-label link embedded in the clean dataset.

One important aspect of trigger is the stealthiness. Patch-based triggers Gu et al. (2017); Liu et al. (2018b) are first proposed but easily detected by a human inspector. Blended Chen et al. (2017), Refool Liu et al. (2020), Wanet Nguyen & Tran (2021) and Ftrojan are proposed to make the triggers

invisible to the human. Under the setting that the attacker can control the training process of the classifier, Nguyen & Tran (2020) utilizes a encoder-decoder neural network to generate the input-aware dynamic trigger, and LIRA Doan et al. (2021b) learns a transform function to produce the invisible triggers. Most attacks are under the dirty-label settings where the label for the poisoned data is not the true label. Turner et al. (2018) imposes stealthiness on the label and defines the clean-label attack as the label being consistent with the input. The clean-label attack is more difficult than the dirty one, since the classifier has to ignore some salient semantic information indicative of the label while establishing the link between the trigger and the label. Zhao et al. (2020) explores the clean-label backdoor attack on video recognition models. All of these attacks can only manipulate the prediction of a given input to one specified target label. Marksman Doan et al. (2022) first studies the multi-label multi-payload backdoor attack which can make the classifier predict any target label given any input. However the attacker in Marksman has a limitation, that it has to be able to control the whole training process of the classifier. In this paper, we consider a scenario with a weaker attacker who can only inject backdoors through data poisoning. Besides, we prove its effectiveness for both dirty- and clean-label attacks, while most backdoor attacks fall short under the clean-label setting.

## 2.2 BACKDOOR DEFENSES

With the rise of backdoor attacks, a variety of backdoor defenses have been proposed to distinguish whether the data or classifier is poisoned Tran et al. (2018); Chen et al. (2022) or to help classifiers remove the embedded backdoors Liu et al. (2018a); Li et al. (2021); Wu & Wang (2021); Chen et al. (2022). STRIP Gao et al. (2019) superimposes various image patterns from different classes to the input and records the entropy of predicted classes for perturbed inputs, where a low entropy implies the presence of a malicious input. It has proven its effectiveness to detect the input-agnostic triggers (e.g., BadNet, Trojan). Spectral signature Tran et al. (2018) relies on the idea that the latent representations will contain a strong signal for the backdoor to detect and remove the poisoned data. Fine-pruning Liu et al. (2018a) analyzes the neuron responses to the clean data and detects the dormant neurons, which are more likely related to the backdoor. It combines pruning and fine-tuning to effectively nullify backdoor attacks. Neural cleanse Wang et al. (2019) utilizes a pattern optimization method and median absolute deviation to detect the presence of a backdoor for one target label in a classifier. Neural attention distillation Li et al. (2021) adopts a teacher classifier to guide the finetuning of the infected student classifier. Adversarial neuron pruning Wu & Wang (2021) prunes sensitive neurons to remove the injected backdoor of an infected classifier. Our experiments prove the PPT as a successful poisoning-based backdoor attack against representative defenses.

## 2.3 ADVERSARIAL PERTURBATIONS

Adversarial perturbations are initially introduced as human-imperceptible and carefully crafted noise added to input data, aiming to catastrophically break the performance during inference Kurakin et al. (2016); Madry et al. (2017); Carlini & Wagner (2017); Croce & Hein (2020). Subsequently, some researchers have utilized adversarial perturbations to promote their work. Adversarial training (AT) Madry et al. (2017); Zhang et al. (2019); Wu et al. (2020), known as the most promising method so far against adversarial attacks, utilizes adversarial examples generated in each training step as data augmentation to gain robustness. Ilyas et al. (2019) demonstrates that adversarial perturbations are directly attributed to the presence of non-robust features derived from patterns in the data distribution that are highly predictive yet incomprehensible to humans. Huang et al. (2021) prevents unauthorized data exploitation by generating error-minimizing perturbations to make the data unlearnable. Fowl et al. (2021) poisons the dataset with adversarial perturbations to degrade the accuracy of the classifier. Adversarial perturbation can also be applied to the backdoor attack. Clean-label backdoor attack Turner et al. (2018) utilizes it to destroy the semantic information in the image for building the link between the pre-defined triggers and the target label for the classification task. Zhao et al. (2020) employs it similarly in video recognition models and generates the universal adversarial perturbations as triggers to activate the backdoor. However, the trigger is identical for one class and patch-based, thus easily detected by human inspection and defenses (e.g., STRIP, spectral signature). Indeed, for classifiers trained without backdoors, black-box adversarial perturbations (i.e., positive triggers) can get a certain attack success rate. Our work stands on this shoulder to show, for the first time, one can achieve a much higher success rate of attacks by poisoning the dataset with positive triggers. Experimental results supporting this claim are presented in supplementary A.1.

## 3 METHODOLOGY

### 3.1 PROBLEMS STATEMENTS

We assume the adversary has access to the clean dataset and the architecture of the classifier to be used by the user. The adversary is allowed to introduce a limited amount of perturbed data to the clean dataset without the permission to interfere with the training process of the classifier. The adversary aims to make classifiers trained on the poisoned dataset perform well on the clean data but predict any specified target class when a corresponding trigger is present.

### 3.2 PRELIMINARIES

**Backdoor Attack**   We focus on poisoning-based backdoor attacks on the image classification task. The adversary can poison $N_p$ samples in the clean dataset to form a poisoned dataset $\hat{D}_p = D_c \cup D_p$, with $D_c = \{(\boldsymbol{x}_i, y_i)\}_i^{N_c}$ and $D_p = \{(\overline{\boldsymbol{x}}_i, \eta(y_i))\}_i^{N_p}$ indicating the clean and poisoned subsets, respectively. $\boldsymbol{x}_i \in \mathbb{X}$, $y_i \in \mathbb{Y}$ denote the clean data and the true label, $\overline{\boldsymbol{x}}_i = \boldsymbol{x}_i \oplus \boldsymbol{\delta}$ is the poisoned data with $\oplus$ being the fusion operator, $\boldsymbol{\delta}$ is the trigger which is related to the $\boldsymbol{x}_i$ and $\eta(y)$, and $\eta$ represents the target labeling function. Clean-label attacks occur when $\eta(y)$ equals to $y$, otherwise they are dirty-label attacks. When users train an infected classifier $f_b$ on the poisoned dataset $\hat{D}_p$, we want to inject backdoors which alter the behavior of $f_b$ so that:

$$\min_{f_b} \frac{1}{N} \sum_{i=1}^{N} \mathcal{L}(f_b(\hat{\boldsymbol{x}}_i), \hat{y}_i), \qquad (\hat{\boldsymbol{x}}_i, \hat{y}_i) \in \hat{D}_p \tag{1}$$
$$f_b(\boldsymbol{x}_i) = y_i, \qquad f_b(\overline{\boldsymbol{x}}_i) = \eta(y_i)$$

We can set $\eta(y_i)$ to any label other than $y_i$ and $N = N_p + N_c$.

**Adversarial perturbation**   Projected gradient descent (PGD) Madry et al. (2017), the most commonly used attack method, formulates generating the adversarial perturbations as a constrained optimization problem. Namely, given a clean input $x^0$, a target label $y$, step size $\alpha$, norm limitation $\epsilon$ and iterations numbers $K$, PGD works as follows:

$$\boldsymbol{x}^{t+1} = \Pi_\epsilon(\boldsymbol{x}^t \pm \alpha \text{sign}(\nabla_{\boldsymbol{x}^t} \mathcal{L}(f(\boldsymbol{x}^t), y))), \quad 0 \le t \le K - 1$$
$$\boldsymbol{\delta} = \boldsymbol{x}^K - \boldsymbol{x}^0, \quad \text{s.t.} \quad \|\boldsymbol{\delta}\|_p \le \epsilon \tag{2}$$

where $\boldsymbol{x}^t$ denotes as the adversarial input at $t$th iteration, $\mathcal{L}(\cdot)$ represents the classification loss (e.g., cross-entropy (CE)), $f$ is a classifier model, and $\Pi_\epsilon$ is the projection onto the $\epsilon$-ball centered at $\boldsymbol{x}^0$. $\delta$ is the adversarial perturbation bounded by an $L_p$-norm, set to $L_\infty$ as default in this paper. Note that the sign between $\boldsymbol{x}^t$ and the backpropagation gradient part can be positive or negative, with positive representing that the generated perturbation moves the input away from the label $y$ (untargeted attack) and vice versa (targeted attack).

**Evaluation Metrics**   We evaluate the performance of the PPT attack adopting two commonly used metrics: accuracy on clean data (ACC) and attack success rate (ASR), i.e., accuracy of predicting poisoned non-target input as the target label. Since we attack any target label at will, we compute the ASR against each class and report its average as ASR.

### 3.3 CATEGORIES OF THE TRIGGERS

Maintaining accuracy without degradation is essential to a successful backdoor attack. On the basis of establishing the input and label links in classifiers trained on the clean dataset, we divide triggers into three categories: positive, neutral, and negative triggers.

Given a benign classifier $f$ trained on the clean dataset, a pair of input and label $(\boldsymbol{x}, y)$, and a target label $\eta(y)$, a trigger $(\boldsymbol{\delta})$ is defined as a positive one for $(\boldsymbol{x}, \eta(y))$ if the $f(\boldsymbol{x} + \boldsymbol{\delta})$ is moved closer to the target label $\eta(y)$, as shown in Fig. 1. We generate the positive triggers following objective:

$$\min_{\|\boldsymbol{\delta}\|_\infty \le \epsilon} \mathcal{L}\left(f\left(\boldsymbol{x} \oplus \boldsymbol{\delta}\right), \eta(y)\right), \tag{3}$$

PGD targeted perturbation is a typical positive trigger. It can induce the input to be categorized into a specified label under certain conditions. The opposite is true for the negative trigger, which will maximize the loss after adding it to the input (e.g., PGD untargeted perturbations). While, neutral triggers do not affect the classifier's prediction, whether they are added to the input or not. In other words, the link between the neutral trigger and the label is independent of the input-label link. Most backdoor triggers belong to the neutral categories. For example, a checkerboard patch (BadNet Gu et al. (2017)) in the lower right corner will not lead the benign classifier to recognize a dog as a cat. It is also true for some input-aware generated triggers. For example, the Marksman Doan et al. (2022), utilizing a trigger function to develop an invisible optimal trigger

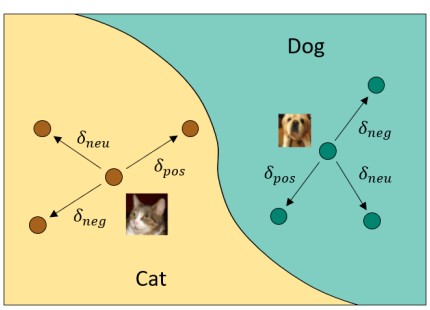

Figure 1: Positive, neutral and negative triggers from a cat image to dog and from a dog image to cat of a benign classifier.

pattern to attack any target label at will, builds a link between the trigger function and the target label during the training phase. This mapping relationship is absent in benign classifiers, meaning that Marksman-generated triggers cannot lead to misclassification (cf. last column in Table 4 in supplementary). Intuitively, the negative trigger is unsuitable for building a link between the trigger and target label since it contains the information pushing it away from the target label. We provide the experimental results with the PGD untargeted perturbations as the triggers in supplementary A.2, suggesting that negative triggers are not an ideal choice. Neutral triggers, which is independent of the input-label link, are the most common choice of the previous backdoor attacks. Instead, we focus on implanting backdoors with positive triggers, which should have less resistance to be classified into target label and are easier to succeed since they take advantage of the input-label links.

### 3.4 POISONING VIA POSITIVE TRIGGERS (PPT)

To inject a backdoor into $f$, we poison a subset of the clean set with a poisoning rate $\rho$ to form a poison dataset $\hat{D}_p = D_c \cup D_p$. The overall framework of the PPT backdoor attack is shown in Fig. 2. We present the details of the PPT algorithm in supplementary A.4.

**Trigger generator**    First, we train a benign classifier on a clean dataset from scratch to minimize the classification loss. Once the training is finished, the benign classifier is fixed as a trigger generator representing the mapping function from the input space to the label space.

**Poison dataset**    By feeding a clean data with a target label $(\boldsymbol{x^0}, \eta(y))$ into the trigger generator, we produce the input-label-aware poisoned data $\boldsymbol{x^K}$ with the targeted PGD according to the Eqn. 2 and replace the clean data $(\boldsymbol{x^0}, y)$ with $(\boldsymbol{x^K}, \eta(y))$. For clean-label attack, the target label $\eta(y)$ equals the true label $y$; otherwise, it is uniformly sampled from the label domain except for the true label. The green and red labels shown in Fig. 2 represent the dirty- and clean-label attacks, respectively.

**Inference**    The backdoors are silently implanted when users train their own classifiers on the poisoned dataset. At the inference stage, given any input and any target label, we can poison the input with a trigger following the Eqn. 2 to manipulate the infected classifier to predict the target label. As shown in Fig. 2, a bird image can be misclassified as a dog or cat with different triggers. Besides, the infected classifiers perform well in the clean data as they predict the bird without the trigger as a bird.

## 4 EXPERIMENTS

### 4.1 EXPERIMENTAL SETTINGS

**Datasets:**    Following the previous backdoor attack papers, we performed comprehensive experiments on four widely-used datasets: SVHN Netzer et al. (2011), CIFAR10 Krizhevsky et al. (2009), GTSRB Stallkamp et al. (2012), and Tiny ImageNet Le & Yang (2015). Note that because of the inconsistent image sizes in the GTSRB, we resize all images to $32 \times 32$ as input. We use various architectures for the classifier $f$: a simple CNN model for SVHN (reported in supplementary A.5),

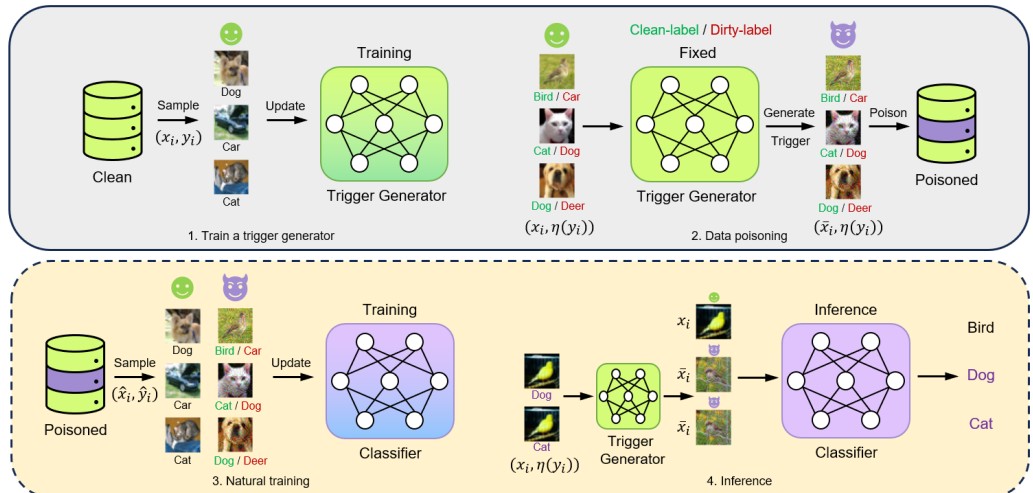

Figure 2: **The Overall framework of the PPT**: The solid box (upper) shows the data poisoning process, and the dashed box (lower) demonstrates the process of training a classifier on the poisoned dataset and performing inference on the clean and poisoned inputs. The triggers of the poisoned data are magnified five times for better understanding.

Pre-activation ResNet18 (PreResNet18) He et al. (2016) for CIFAR10 and GTSRB, and ResNet18 He et al. (2016) for Tiny ImageNet as suggested by Doan et al. (2022).

**Hyperparameters:** We train the trigger generator and classifier for 300 epochs with a batch size of 128 utilizing a SGD optimizer with the Cross-Entropy (CE) loss. The initial learning rate was set to $1 \times 10^{-2}$, which decayed to one-tenth after 100 and 200 epochs, respectively. Following the settings of Marksman, the maximum $l_\infty$ norm-bounded perturbation $\epsilon$ was set to 0.05 for all datasets. All experiments were conducted on one Nvidia RTX 3090 GPU with 24GB CUDA memory.

**Configurations:** To our best knowledge, this paper is the first work that explores the multi-trigger and multi-payload poisoning-based backdoor attack, which enables classifying inputs poisoned by input-label-aware triggers into any target class without interfering with the training process of the classifier. Classifiers trained on this poisoned dataset are implanted with multiple backdoors so that we can manipulate the input to fall into any target class. Following the settings of Marksman Doan et al. (2022), we implement three typical baseline methods: **PatchMT**, **WaNetMT**, and **Marksman** for comparison. Details of these baselines are in supplementary A.3.

## 4.2 ATTACK PERFORMANCE

We consider both dirty- and clean-label settings for data poisoning. Since the data and label of the dirty-label setting are inconsistent, a lower poisoning rate is better for evading human random inspection. The clean-label one does not change the input's label, making it harder to inject the backdoors because the classifier needs to ignore information in the data to establish a link between the trigger and the label. While ensuring stealthiness, we can add more clean-label poisoned data without worrying about human spot checks. For each clean test input $(\boldsymbol{x}, y)$, we enumerate all target labels except $y$ to generate triggers $\boldsymbol{\delta}$ and feed $\boldsymbol{x} \oplus \boldsymbol{\delta}$ into the classifier to check the ASR. The attack is considered successful if the classifier classifies $\boldsymbol{x} \oplus \boldsymbol{\delta}$ as the target label. The ASR reported in this paper is the average of ASR of each target label. We provide the accuracy and ASR with $1\%$ and $10\%$ poisoning rates for dirty- and clean-label in Table 1, respectively. More experimental results with different poisoning rates are provided in supplementary A.6, Our method outperforms other poisoning-based multi-label and multi-payload backdoor attacks with a significantly higher ASR.

The ACC and ASR of three typical attacks and PPT are presented in Table 1. For the dirty-label attack, the input-aware triggers generated by WaNetMT and Marksman are unable to form an effective attack with a small (viz. $1\%$) poisoning rate, while the patch-based PatchMT has a better ASR

Table 1: Accuracy (%) and ASR(%) across four datasets. Benign represents the accuracy of the classifier trained on the clean dataset. Bold values indicate the best ASR.

| Dataset | Benign ACC | PatchMT ACC | PatchMT ASR | WaNetMT ACC | WaNetMT ASR | Marksman ACC | Marksman ASR | PPT ACC | PPT ASR |
|---|---|---|---|---|---|---|---|---|---|
| Dirty-label $\eta(y) \neq y$ poisoning rate = 1% | | | | | | | | | |
| SVHN | 95.21 | 94.66 | **99.32** | 94.55 | 1.22 | 94.75 | 0.92 | 94.79 | 93.13 |
| CIFAR-10 | 94.32 | 93.90 | 94.71 | 94.42 | 5.62 | 93.62 | 0.89 | 94.22 | **98.53** |
| GTSRB | 99.17 | 98.60 | 23.6 | 98.92 | 0.60 | 99.08 | 10.82 | 99.34 | **85.48** |
| Tiny-ImageNet | 59.02 | 58.32 | 0.23 | 58.89 | 0.36 | 58.97 | 0.68 | 59.38 | **22.23** |
| Clean-label $\eta(y) = y$ poisoning rate = 10% | | | | | | | | | |
| SVHN | 95.21 | 94.18 | 46.28 | 94.91 | 0.71 | 95.17 | 0.58 | 94.30 | **86.83** |
| CIFAR-10 | 94.32 | 94.20 | 10.84 | 94.28 | 0.91 | 94.72 | 0.62 | 93.92 | **92.22** |
| GTSRB | 99.17 | 99.18 | 0.03 | 99.13 | 0.03 | 98.86 | 0.02 | 99.06 | **39.83** |
| Tiny-ImageNet | 59.02 | 58.89 | 0.83 | 58.67 | 0.21 | 58.31 | 0.24 | 58.96 | **24.01** |

performance. It indicates that without controlling the classifier training process, it is far simpler to let the DNN memorize multiple definite patterns than recognize the trigger functions applied to the input. In contrast, the PPT attack can maintain a high ASR across four datasets. As mentioned before, when the number of classes increases, it becomes more difficult to simultaneously implant an equal number of backdoors within the classifier. The ASR of PatchMT drops from $84.38\%$ for 10 classes in CIFAR10 to $23.6\%$ for 43 classes in GTSRB and to $0.23\%$ similar to a random guess for 200 classes in Tiny ImageNet. Since the proposed PPT is dependent on the input-label link, it can be easily embedded in the classifier, proved by the $84.97\%$ ASR of GTSRB and $24.53\%$ ASR of Tiny ImageNet, a significant improvement compared to the PatchMT. Besides, the accuracy of PPT is superior to the other three baselines and similar to the benign model, which suggests that the positive triggers contain features that can facilitate the image classification, consistent with the opinions in Ilyas et al. (2019). Regarding the clean-label attack, all three baselines fail to inject the backdoors inside the classifiers with a $10\%$ poisoning rate. Since the input contains abundant semantic information related to the true label, it prevents the classifier from building a solid link between the trigger and the target label. In comparison, the PPT is a highly effective attack that achieves a high ASR with negligible ACC degradation across four datasets. More experimental results with $10\%$ and $50\%$ poisoning rates for dirty- and clean-label attacks are provided in Table 6 in supplementary. With an increase in the poisoning rate, all backdoor attacks can improve their ASR with a higher risk of being detected. Nevertheless, all three baselines fail to form a successful backdoor attack for GTSRB and Tiny ImageNet datasets under the clean-label settings.

## 4.3 ABLATION STUDIES

**Poisoning Rate** We evaluate the ACC and ASR of different poisoning rate (i.e.,$|D_p|/(|D_p|+|D_c|)$) within a reasonable range. The results of dirty-label attack with poisoning rate ranging from $1\%$ to $10\%$ are shown in Fig. 3. As the poisoning rate increases, the ASR of the attack increases, while the ACC only drops by a negligible amount. A small amount of poisoned data already guarantees a high ASR for datasets with ten classes, such as SVHN, CIFAR10. For datasets with more classes, the increase in the poisoning rate provides much of a performance boost. In the case of the GTSRB, the Attack Success Rate (ASR) improves by $9.64\%$ with a slight decrease in accuracy of $0.40\%$. Similarly, for the Tiny ImageNet, the ASR improves by a significant $51.29\%$ with a reduction in accuracy of $1.54\%$. More results of the clean-label attack are provided in Fig. 9 in supplementary.

**Trigger Generator & Classifier** We explore how the architecture of the trigger generator and the classifier affects the performance of the attack. Compared to the assumption that the trigger generator and classifier employ the same architecture, a more general setting is that the attacker does not know the classifier structure adopted by the user. We use four typical convolutional neural networks (CNNs) as trigger generators and classifiers to test the dirty-label attack's accuracy and success rate with $1\%$ poisoning rate. As shown in Table 2, ResNet18, used as a trigger generator, demonstrates a high ASR regardless of the architecture used for the classifier. It indicates that consistency in the architecture

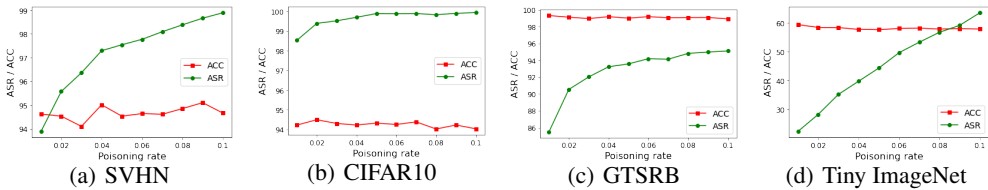

(a) SVHN  (b) CIFAR10  (c) GTSRB  (d) Tiny ImageNet

Figure 3: Ablation studies of poisoning rate on (a) SVHN, (b) CIFAR10, (c) GTSRB, and (d) Tiny ImageNet for dirty-label attack.

of generators and classifiers is *not* necessary for effective backdoor attacks. As long as the neural network can establish a link between input and label, it can serve as a functional trigger generator. In supplementary, clean-label attack yields similar results, depicted in Table 7.

Table 2: Accuracy (%) and ASR(%) of dirty-label attack for different model architectures on CIFAR10. Bold values represent the best ASR for each classifier.

| ACC (%) / ASR (%) | Trigger Generator | | | |
|---|---|---|---|---|
| | EfficientNet-B0 | MobileNet-V2 | ResNet18 | PreResNet18 |
| **Classifier** EfficientNet-B0 | 91.62 / **95.61** | 92.23 / 87.32 | 91.94 / 93.52 | 92.03 / 91.23 |
| MobileNet-V2 | 94.02 / 89.01 | 93.82 / 93.98 | 94.11 / **96.89** | 93.92 / 95.62 |
| ResNet18 | 94.28 / 88.17 | 94.32 / 92.21 | 94.02 / **99.15** | 94.11 / 98.28 |
| PreResNet18 | 94.32 / 88.23 | 94.41 / 91.59 | 94.31 / **98.58** | 94.22 / 98.53 |

## 4.4 DEFENCE PERFORMANCE

Here we evaluate the infected classifier trained on the proposed poisoned dataset against the popular defense mechanisms, including STRIP Gao et al. (2019), Spectral Signature Tran et al. (2018), and Fine-Pruning Liu et al. (2018a). To save space, more evaluation results against Neural Cleanse Wang et al. (2019) and two post-training defenses called Neural Attention Distillation (NAD) Li et al. (2021) and Adversarial Neuron Pruning (ANP) Wu & Wang (2021) are provided in supplementary A.8. Unless otherwise stated, the results are for the dirty-label attack with 1% poisoning rate. Experimental results for the clean-label attack with 10% poisoning rate are presented in supplementary A.8.

**STRIP** is a typical testing-time defense method. It superimposes different images to the input and records the entropy of predicted classes, where a low entropy implies the presence of a malicious input. We plot the entropy of the clean and poisoned inputs in Figs. 4 (also Fig. 11 in supplementary). Similar energy distribution of the clean and poisoned inputs indicates defense failure.

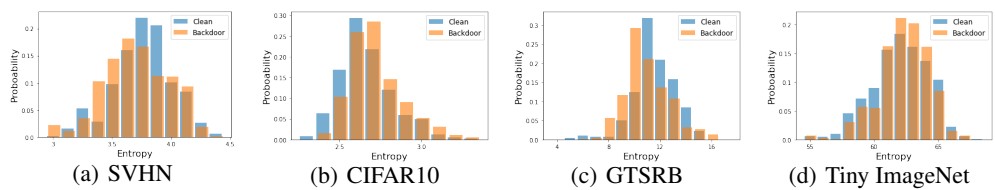

(a) SVHN  (b) CIFAR10  (c) GTSRB  (d) Tiny ImageNet

Figure 4: Entropy distributions on (a) SVHN, (b) CIFAR10, (c) GTSRB, and (d) Tiny ImageNet.

**Spectral Signatures** computes the top singular vector of the covariance matrix of the latent representations using a subset of clean data and calculates the correlation of each input to this top singular vector. The input with an outlier score is recognized as poisoned data. For each dataset, we randomly select a target label and compute the correlation score for both the clean and poisoned inputs. As shown in Figs. 5 (also Fig. 12 in supplementary), there is no clear distinction between clean and poisoned inputs, which verifies the stealthiness of the PPT attack.

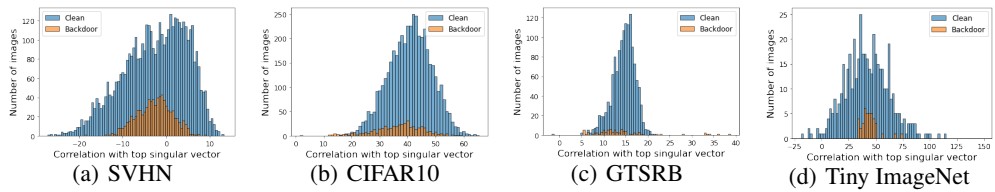

Figure 5: Spectral signature on (a) SVHN, (b) CIFAR10, (c) GTSRB, and (d) Tiny ImageNet.

**Fine-Pruning** assumes the dormant neurons are more likely to tie to the backdoor. Given a special layer, it detects and gradually prunes the dormant neurons with low activations to mitigate the backdoor. Following previous works Nguyen & Tran (2021); Doan et al. (2022), we prune the last CNN layer and provide the results in Fig. 6 (also Fig. 13 in supplementary). ACC and ASR have the same trend, indicating that reducing ASR without degrading accuracy is impossible.

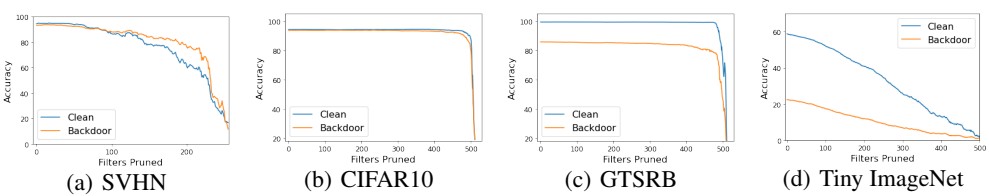

Figure 6: Fine-pruning on (a) SVHN, (b) CIFAR10, (c) GTSRB, and (d) Tiny ImageNet

### 4.5 VISUALIZATION OF POISONED DATA

**GradCam** Selvaraju et al. (2017) We provide the visualization of the clean and poisoned data in Fig. 7. Based on the true labels, the GradCam activations of the clean data computed by a benign classifier and the activations of the poisoned one computed by an infected classifier are included. The heatmap of the clean input looks like the poisoned one, suggesting the infected classifier extracts information from similar regions as the benign one.

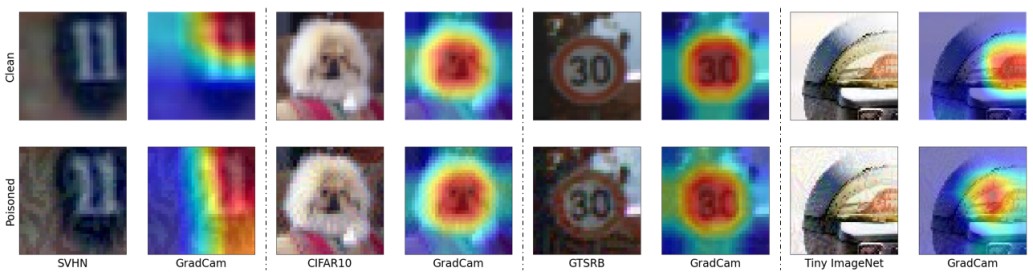

Figure 7: GradCam visualization between the clean (top) and poisoned images (bottom) computed by a clean classifier and an infected one across four datasets.

## 5 CONCLUSION

This paper introduces a new categorization of backdoor triggers based on whether they can move the input close to the target label and achieve a high attack success rate (ASR) in multi-label and multi-payload backdoor attacks by poisoning the dataset with positive triggers for both dirty- and clean-label attacks. Additionally, we demonstrate the resilience of the proposed PPT attack against several popular backdoor defenses. This research opens up new possibilities for designing backdoor triggers and encourages future exploration in defense mechanisms.

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

# A APPENDIX

## A.1 TARGETED ADVERSARIAL ATTACK OR BACKDOOR ATTACK

Since the generation of the positive trigger is similar to the targeted adversarial perturbation under the black-box attack settings, one may question whether the high attack success rate relies entirely on the targeted adversarial perturbations rather than on backdoors. We evaluate the attack success rate of targeted adversarial perturbations PGD-10 (i.e. positive triggers) of benign classifier trained on the clean dataset and infected classifier trained on the poisoned dataset with $1\%$ poisoning rate. We take PreResNet18 as a default classifier and choose several common neural networks as the surrogate models (i.e., trigger generators) to compute the attack success rate.

| classifier | EfficientNet-B0 | MobileNet-V2 | PreResNet18 | ResNet18 |
|---|---|---|---|---|
| Benign | 40.89 | 43.80 | 75.52 | 61.36 |
| Infected | 81.57 | 84.06 | 96.52 | 96.82 |

Table 3: Attack success rate $(\%)$ of the adversarial perturbation constrained with $l_\infty = 8/255$ for the benign and infected classifiers.

As the attack success rate shown in Table 3, targeted adversarial perturbations have a limited attack success rate on benign classifier. Our work stands on this shoulder by poisoning the dataset with positive triggers to achieve a much higher attack success rate ($34.35\%$ increase on average).

## A.2 ABLATION STUDIES FOR TRIGGER TYPE

In this section, we compare the effect of different triggers on the ASR. Given a benign classifier $f$ trained on the clean dataset, a pair of input and label $(\boldsymbol{x}, y)$, and a target label $\eta(y)$. We generate the positive trigger $\delta_p$ and negative trigger $\delta_n$ as follows:

$$\min_{\|\boldsymbol{\delta_p}\|_\infty \leq \epsilon} \mathcal{L}\left(f\left(\boldsymbol{x} \oplus \boldsymbol{\delta_p}\right), \eta(y)\right),\tag{4}$$

$$\max_{\|\boldsymbol{\delta_n}\|_\infty \leq \epsilon} \mathcal{L}\left(f\left(\boldsymbol{x} \oplus \boldsymbol{\delta_n}\right), \eta(y)\right),\tag{5}$$

For neutral triggers, we adopt the patch-based triggers in BadNets. We first train two benign PreResNet18 with different initializations on the clean dataset, one as the trigger generator and another for evaluating the ASR of various triggers. For each type of trigger, we poison one percent of the dataset and train an infected classifier on the poisoned dataset. Then, we evaluate the ASR of each trigger on the benign and infected classifiers. The experiment is conducted on the CIFAR10 dataset under the dirty-label all2one attack setting. The target label is randomly selected from all possible labels (set to 7 in this section). As shown in Table 4, the positive trigger has a high ASR on the benign classifier, a property we want to exploit to improve the ASR in this paper. The negative trigger is not an ideal choice as it cannot form an effective attack in benign and infected classifiers. As for the neutral trigger, it doesn't influence the output of the benign classifier, but will lead to a wrong prediction when the backdoor is present in the infected classifier. Besides, Marksman belongs to the neutral trigger, which has little impact on the prediction of the benign classifier.

| Classifier | Positive | Negative | BadNets | Marksman |
|---|---|---|---|---|
| Benign | 74.82 | 0.00 | 0.52 | 0.63 |
| Infected | 99.91 | 0.01 | 99.87 | 32.77 |

Table 4: Attack success rate $(\%)$ of different triggers on benign and infected classifiers for the dirty-label all2one attack.

## A.3 DETAILS OF THE BASELINES

This section elaborates on the details of how the three baseline methods construct the poisoned datasets. We design the trigger parameters for each class in the dataset and poison a certain proportion

of clean data by adding different trigger patterns repeatedly for different target classes. **PatchMT** is an extension of the patch-based BadNets Gu et al. (2017), where we use different patch locations and Black-and-white blocks for different target classes. Similarly, **WaNetMT** is an extension of WaNet Nguyen & Tran (2021), where we select different combinations of the grid size $k$ (between 4 and 8) and the warping strength $s$ (between 0.5 and 0.75) of warping fields for various target classes. For **Marksman**, we follow the training process in the original paper with the poisoning rate set as 10% and utilize the well-trained trigger function to generate the triggers. The PatchMT's trigger is only label-aware, while the WaNetMT and Marksman's triggers are input-label-aware and invisible. We visualize the poisoned images in Fig 8.

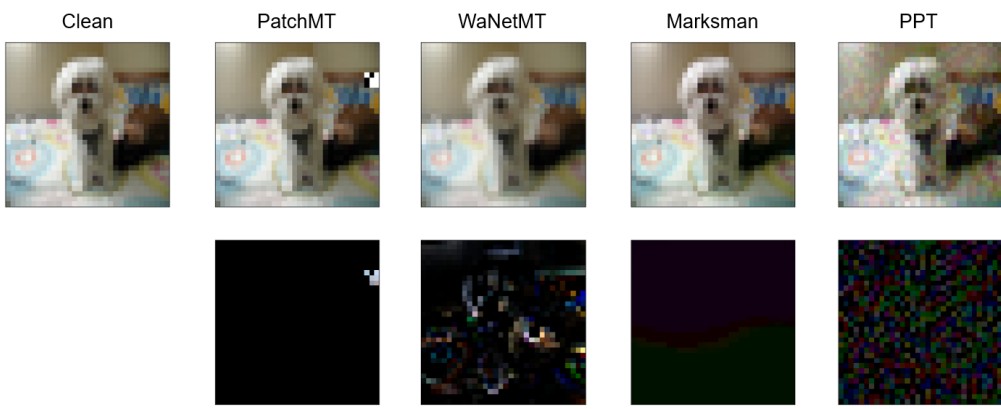

Figure 8: Clean and poisoned images (top) of four backdoor attacks. The trigger pattern (bottom) are magnified five times for better visualization.

Once the poisoned dataset is finalized, only a clean or poisoned version of each data will be fed into the classifier for training. It is significantly different from the original settings of WaNet and Marksman, which control the training process and poison a certain percentage of the data in the mini-batch. Assuming that the poisoning rate is 10%, each input has a 10% probability of not being poisoned within each epoch. Following 100 epochs of training, the classifier has a $(0.9)^{100} = 2.7 \times 10^{-5}\%$ chance of solely using its clean version for each sample. This indicates that the classifier can learn from both the clean and poisoned versions of all data in WaNet and Marksman.

### A.4 ALGORITHM FOR POISONED DATASET

---

**Algorithm 1** The proposed PPT to generate the poisoned dataset by enhancing the positive triggers implicitly embedded in the mapping from the input space to the label space.

---

**Input:** Clean dataset $D = \{(x_i, y_i), i = 1, ..., N\}$, Trigger generator $G$ with parameters $\theta_G$, Loss function $\mathcal{L}$, Poisoning rate $\rho$, Learning rate $\gamma$, training iterations $K_G$
**Output:** A poisoned dataset
**Step one: Train model $G$ on dataset $D$**
    **Repeat**
        Initial $i \leftarrow 0$, sample mini-batch $D_m = \{(x, y)\} \subset D$;
        Update $\theta_G$: $\theta_G \leftarrow \theta_G - \gamma \nabla_{\theta_G} \sum_{D_m} \mathcal{L}(G(x), y), i \leftarrow i + 1$;
    **Until** $i = K_G$
**Step two: Generate the poisoned dataset**
    Initial $t \leftarrow 0$, sample $D_p = \{(x^0, y)\} \subset D, D_c = D \setminus D_p, \quad s.t. |D_p| = |D| \times \rho\%$
    **Repeat**
        $x^{t+1} = \Pi_\epsilon(x^t + \alpha \text{sign}(\nabla_{x^t} \mathcal{L}(G(x^t), \eta(y)))), \quad \forall x^0 \in D_p$
    **Until** $t = K$
    Replace all $(x^0, y)$ in $D_p$ with $\{(x^K, \eta(y))\}$
    Get the poisoned dataset $\hat{D}_p = D_p \cup D_c$.

---

## A.5  ARCHITECTURE FOR SVHN

We apply a convolution neural networks for SVHN dataset, shown in Table 5

| Layer | Filter | Filter Size | Stride | Padding | Activation |
|---|---|---|---|---|---|
| *Conv2d* | 32 | $3 \times 3$ | 1 | 1 | ReLU |
| *Conv2d* | 32 | $3 \times 3$ | 1 | 1 | ReLU |
| *MaxPool2D* | - | $2 \times 2$ | 2 | 0 | - |
| *Conv2d* | 64 | $3 \times 3$ | 1 | 1 | ReLU |
| *Conv2d* | 64 | $3 \times 3$ | 1 | 1 | ReLU |
| *MaxPool2D* | - | $2 \times 2$ | 2 | 0 | - |
| *Conv2d* | 128 | $3 \times 3$ | 1 | 1 | ReLU |
| *Conv2d* | 128 | $3 \times 3$ | 1 | 1 | ReLU |
| *MaxPool2D* | - | $2 \times 2$ | 2 | 0 | - |
| *Conv2d* | 256 | $3 \times 3$ | 1 | 1 | ReLU |
| *Conv2d* | 256 | $3 \times 3$ | 1 | 0 | ReLU |
| *MaxPool2D* | - | $2 \times 2$ | 2 | 0 | - |
| *Linear* | 10 | - | - | - | Softmax |

Table 5: Detailed architecture of SVHN classifier. Each *Conv2d* layer is followed by a Batch Normalization and a Dropout ($p = 0.3$) layer.

## A.6  PERFORMANCE WITH A HIGH POISONING RATE

We provide more experimental results with higher poisoning rates, $10\%$ and $50\%$ for dirty- and clean-label attacks, respectively. For the dirty-label attack, the proposed PPT achieves the best ASR on the three datasets and performs similarly to PatchMT on SVHN without the risk of being detected. Besides, the PPT attack is the only one that can achieve a high ASR of GTSRB and Tiny ImageNet for the clean-label attack.

Table 6: Accuracy ($\%$) and ASR($\%$) across four datasets. Benign represents the accuracy of the classifier trained on the clean dataset.

| Dataset | Benign ACC | PatchMT ACC | ASR | WaNetMT ACC | ASR | Marksman ACC | ASR | ours ACC | ASR |
|---|---|---|---|---|---|---|---|---|---|
| Dirty-label $\eta(y) \neq y$ poisoning rate = $10\%$ | | | | | | | | | |
| SVHN | 95.21 | 93.97 | **100** | 90.62 | 95.46 | 90.81 | 50.43 | 94.66 | 98.89 |
| CIFAR-10 | 94.32 | 92.69 | 99.60 | 93.30 | 90.32 | 92.25 | 37.52 | 94.07 | **99.92** |
| GTSRB | 99.17 | 97.92 | 94.25 | 98.40 | 82.84 | 98.19 | 32.52 | 98.94 | **95.12** |
| Tiny-ImageNet | 59.02 | 53.28 | 5.03 | 54.48 | 50.40 | 55.86 | 4.09 | 57.84 | **63.52** |
| Clean-label $\eta(y) = y$ poisoning rate = $50\%$ | | | | | | | | | |
| SVHN | 95.21 | 94.17 | 86.72 | 94.28 | 0.98 | 94.33 | 2.75 | 93.72 | **87.98** |
| CIFAR-10 | 94.32 | 93.75 | 23.51 | 93.53 | 1.14 | 93.75 | 1.43 | 92.57 | **96.68** |
| GTSRB | 99.17 | 99.20 | 0.03 | 99.15 | 0.01 | 99.18 | 0.02 | 98.89 | **42.89** |
| Tiny-ImageNet | 59.02 | 57.71 | 9.52 | 57.40 | 0.35 | 55.59 | 2.78 | 57.53 | **55.57** |

## A.7  ABLATION STUDIES FOR CLEAN-LABEL ATTACK

We provide the experimental results of clean-label attack for ablation studies.

**Poisoning Rate**  As shown in Fig 9, the clean-label attack demonstrates similar results as the dirty-label attack. As the poisoning rate increases, the ASR of the attack increases, while the ACC only drops by a negligible amount.

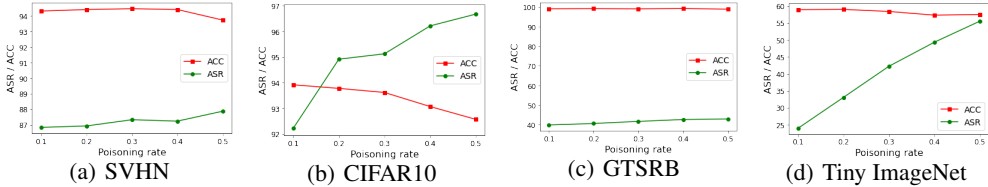

(a) SVHN  (b) CIFAR10  (c) GTSRB  (d) Tiny ImageNet

Figure 9: Ablation studies of poisoning rate on (a) SVHN, (b) CIFAR10, (c) GTSRB, and (d) Tiny ImageNet for dirty-label attack.

**Generator & Classifier**  The experimental results in Table 7 reaffirm that an attacker can effectively execute the poisoning-based attack without prior knowledge of the user's classifier.

Table 7: Accuracy (%) and ASR(%) of clean-label attack for different model architectures on CIFAR10. Bold value represents the best ASR for each classifier.

| ACC (%) / ASR (%) | Trigger Generator | | | |
|---|---|---|---|---|
| | EfficientNet-B0 | MobileNet-V2 | ResNet18 | PreResNet18 |
| EfficientNet-B0 | 91.30  85.68 | 91.95 / 74.27 | 91.50 / **86.17** | 91.54 / 78.01 |
| MobileNet-V2 | 93.76 / 68.67 | 93.56 / 81.18 | 93.51 / **86.63** | 93.54 / 82.59 |
| ResNet18 4 | 93.88 / 60.70 | 94.04 / 75.44 | 94.08 / **95.63** | 94.17 / 92.36 |
| PreResNet18 | 93.95 / 60.37 | 94.01 / 74.86 | 93.98 / **95.07** | 93.92 / **92.22** |

(The left-hand side of the table is labeled "Classifier" vertically.)

## A.8 MORE EXPERIMENTAL RESULTS AGAINST DEFENSES

**Neural cleanse**  is an effective defense to detect the backdoored classifier based on pattern optimization approaches. For each possible label, it derives a reverse optimal pattern using the gradient backpropagation method that causes clean data to be classified into the target label. Then, it utilizes the median absolute deviation to compute an anomaly index, which indicates a backdoored classifier when it exceeds a threshold of 2. However, it assumes that the backdoor trigger is patch-based and only one label is attacked, suggesting that it is unsuitable to detect the backdoor attack proposed in this paper. We provide the results of neural cleanse across three datasets in Fig. 10.

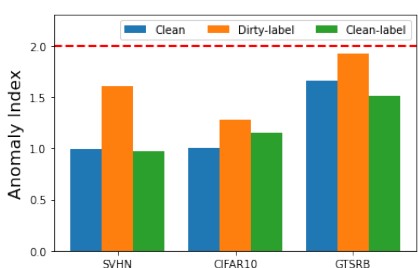

Figure 10: Neural cleanse.

**NAD & ANP**  We evaluate the attack performance against two typical post-training defenses: Neural Attention Distillation (NAD) and Adversarial Neuron Pruning (ANP). We follow the same settings of NAD and ANP, which utilize a small subset of clean data (5%) to remove the backdoor. The accuracy and ASR of the infected models before and after they undergo the erasing processes of NAD and ANP are shown in Table 8. For NAD and ANP, the distillation or pruning drastically reduces accuracy and attack success rate synchronously, indicating that they are ineffective against the PPT.

| Attack | Before | | NAD | | ANP | |
|---|---|---|---|---|---|---|
| | ACC | ASR | ACC | ASR | ACC | ASR |
| dirty-label | 94.22 | 98.53 | 38.76 | 7.19 | 67.41 | 28.45 |
| clean-label | 93.92 | 92.22 | 35.29 | 7.08 | 60.35 | 21.83 |

Table 8: Performance against NAD and ANP.

**STRIP for clean-label attack**    The energy distribution of the clean and poisoned inputs is similar to each other shown in Fig 11. we cannot distinguish whether the input is poisoned by STRIP.

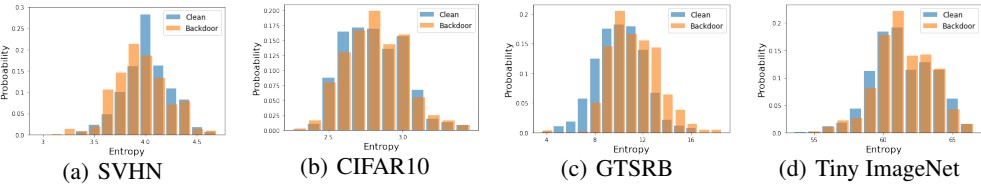

Figure 11: Entropy distributions on (a) SVHN, (b) CIFAR10, (c) GTSRB, and (d) Tiny ImageNet for clean-label attack.

**Spectral Signatures for clean-label attack**    We cannot distinguish the poisoned inputs using the spectral signature since there is no outlier correlation score for the clean-label attack, shown in Fig 12.

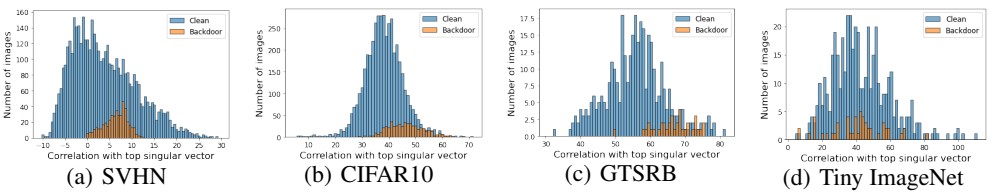

Figure 12: Spectral signature on (a) SVHN, (b) CIFAR10, (c) GTSRB, and (d) Tiny ImageNet for clean-label attack.

**Fine-pruning for clean-label attack**    As shown in Fig 13, when the ASR decreases, the accuracy of the classifier also decreases. This indicates that fine-pruning cannot maintain accuracy while reducing the attack success rate.

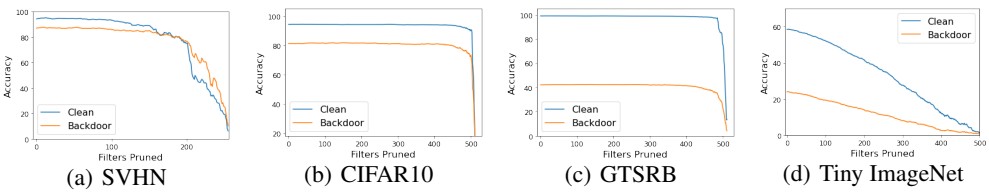

Figure 13: Fine-pruning on (a) SVHN, (b) CIFAR10, (c) GTSRB, and (d) Tiny ImageNet for clean-label attack.

