# OpenReview forum: "Poisoning-based Backdoor Attacks for Arbitrary Target Label with Positive Triggers"
_ICLR.cc/2024/Conference — ICLR 2024 Conference Withdrawn Submission_

### Official Review · Reviewer_N6ns · 2023-10-20

**Soundness:** 3 good
**Presentation:** 3 good
**Contribution:** 2 fair
**Rating:** 3
**Confidence:** 4

**Summary:**

This paper introduced a new way to classify backdoor triggers based on how they affect the distance between the malicious inputs and the target label. Based on this categorization, the authors leverage the triggers that can move the inputs closer to the target label, called positive triggers, to achieve high ASR in the multi-label and multi-payload backdoor attack.  Unlike previous work, this method can effectively attack the victim models without controlling the training process, under both dirty-label and clean-label settings.

**Strengths:**

1. The idea of categorizing backdoor triggers based on whether they can move the inputs close to target label is quite interesting and could be useful for further backdoor studies.
2. The attack effectiveness seems superior compared to other baselines. Especially, the attack can achieve high ASRs given extremely low poisoning rate (1%) in the dirty-label settings.
3. The experiments are quite extensive, testing with various datasets, model architectures, and backdoor defenses.
4. The writing of the paper is quite clear and easy to follow.

**Weaknesses:**

1. My main concern is that whether using "positive triggers", which is similar to adversarial perturbations, is relevant to the goal of backdoor attack by data poisoning. While poisoning can improve the ASRs, the trigger generator alone can work as a functional adversarial examples generator, as the results shown in Table 3 in section A.1 indicate. For this reason, the success of this attack with low poisoning rates (1% in dirty-label and 10% in clean-label settings) might come from the fact that the attack relies on the effectiveness of the adversarial perturbations and poisoning has only limited impact. If so, the performance of the attack is not very surprising, and the attack might be mitigated by adversarial defenses. While this might not necessarily be the case, I think experiments against standard adversarial defenses, such as adversarial training, are needed.
2. I appreciate that the experiments are conducted with various datasets to show the attack's effectiveness, however, the performances on GTSRB in clean-label settings and Tiny-ImageNet in both settings are quite underwhelming and hard to be considered successful.
3. While this method seems to outperform other baselines, the generated triggers are much more visible, which might expose the malicious inputs to human inspection.
4. (merely minor) typo: "dirty-label" in the caption of Figure 9 should be "clean-label"

**Questions:**

1. Regarding my main concern above, I suggest the authors consider additional experiments against adversarial defenses, e.g., adversarial training.
2. Although the paper already includes experiments with 6 backdoor defenses, some of them are quite old and can hardly be considered state-of-the-art anymore (Neural Cleanse and Spectral Signatures were introduced in 2019, Fine-pruning and STRIP were introduced in 2018). It would better show the attack's robustness if there are more evaluations against more recent backdoor defenses. I would recommend [1] and [2], especially [1] which seems to be highly effective in detecting poisoned samples.
3. All datasets used in this work have relatively low resolution. Could this method also work with high resolution datasets, such as CelebA, PubFig, ImageNet?

[1] Zeng, Y., Park, W., Mao, Z. M., & Jia, R. " Rethinking the backdoor attacks' triggers: A frequency perspective." ICCV 2021.
[2] Zheng, R., Tang, R., Li, J., & Liu, L. "Data-free backdoor removal based on channel lipschitzness." ECCV 2022.

---

### Official Review · Reviewer_GFty · 2023-10-29

**Soundness:** 2 fair
**Presentation:** 3 good
**Contribution:** 2 fair
**Rating:** 3
**Confidence:** 4

**Summary:**

This paper introduces a new data poisoning attack that can cause any input to be misclassified as any target label. It leverages adversarial perturbations as the backdoor trigger to achieve the attack goal. Specifically, this paper trains a surrogate model using the same clean training dataset and model architecture as the victim model. It then employs the adversarial attack PGD to generate adversarial examples as the poisoning data to inject into the clean training set. During inference, the attack generates adversarial examples on the surrogate model to cause misclassification on the victim model. The experiments were conducted on four image datasets and several model architectures. The results demonstrate that the proposed poisoning attack can achieve a high attack success rate without affecting the clean accuracy.

**Strengths:**

1. It is an important and timely topic for studying poisoning attacks against deep neural networks.

2. The paper is well-written and clearly explains the threat model and the proposed attack procedure.

**Weaknesses:**

1. It is interesting to see a poisoning attack that can cause any input to be misclassified to any target label. However, after reading Section 3.4, it appears that the proposed technique is simply a transferable adversarial attack. This paper assumes knowledge of the entire training dataset and the model architecture. During inference, it employs PGD to generate adversarial examples as trigger-injected inputs. It is not clear why this is a data poisoning attack. Existing works have already demonstrated the transferability of adversarial examples. There is no need to poison the training set to achieve the attack goal.

2. The ablation study in Figure 3 does not seem reasonable. Since the proposed attack is essentially a transferable adversarial attack, it should exhibit a high attack success rate even with a zero poisoning rate. The results do not make sense. If this is indeed the case, then one can simply use multiple surrogate models to generate more robust adversarial examples, a technique that has been extensively studied in the literature over the past decade.

3. As the proposed technique is more aligned with adversarial attacks rather than poisoning attacks, the countermeasures should focus on techniques specifically designed for adversarial attacks, such as adversarial training and randomized smoothing. The current evaluation of defenses does not adequately demonstrate the effectiveness of the attack.

**Questions:**

N/A

---

### Official Review · Reviewer_dCfv · 2023-10-31

**Soundness:** 3 good
**Presentation:** 3 good
**Contribution:** 3 good
**Rating:** 5
**Confidence:** 5

**Summary:**

The authors propose a multi-label and multi-payload backdoor attack that relies on data poisoning to target any class of the poisoned model at inference time. They assume knowledge of the entire training dataset, all hyperparameters, and the defender's model architecture to generate poisoned samples. In a poison-label setting, the authors show that their attack succeeds with only 1% of poisoned samples injected into the defender's training dataset.

**Strengths:**

Overall, I found the paper interesting, but I have some open questions. Here are the strengths.

**Use-Case.** The studied problem of targeting multiple classes is interesting and has yet to be thoroughly explored. Especially with the advance of more capable models that classify into thousands of classes, I can see the relevance of these types of attacks in the future.

**Well written.** The paper was well-written and easy to follow for most parts.

**Evaluation.** I appreciate that the authors included many defenses and ablation studies in their paper.

**Weaknesses:**

**Threat Model.**
I am missing a concise threat model. Could the authors please explain how they measure ASR exactly? Do you test ASR for every class equally? Please summarize your attacker's capabilities and goals (since you propose a new type of attack). Can the attacker choose what samples to poison, or are they restricted in their choice in some way (e.g., some classes are inaccessible)? What capabilities and goals does the defender have?

**Baseline ASR.** Since your poison triggers are adversarial examples, I wonder what the baseline ASR would be if you did not inject _any_ poisoned sample. Figure 3 shows that injecting $1$% poisoned samples already achieves $98.5$% on CIFAR-10. What is the ASR for $0$% poisoned samples?

**Detectability.** The poisoned samples in Figure 7 have clearly visible noise patterns imprinted on them. The authors state that epsilon was fixed to $0.05$ (I assume this is out of $1$), translating to about $\epsilon=12/255$. Clearly, perturbations at such epsilon are easily visually perceptible. Is there a trade-off between $\epsilon$ and the efficiency of your poisoning attack (i.e., the number of samples needed to achieve a given ASR)? If yes, please include that in your response.

**Robustness.** How robust is your attack? I would love to see what conditions must be met to remove the backdoor. What if the defender can access a lot of clean training data to remove the backdoor? Since your attack relies on adversarial examples, what if the defender used adaptive attacks such as adversarial training?

**Minor Comments**

* In Eq. 1, I assume the minimization is over the parameters of the poisoned model
* Fig. 7 is unclear to me. I am assuming the ACC is depicted for both clean and poisoned models. Please also include the ASR and label it clearly.
* The concept of "Positive Triggers (PPT)" was unclear to me until very late in the paper.

**Questions:**

How robust is your attack against adaptive defenses, e.g., when the defender trains their model with adversarial training?

What is the trade-off between $epsilon$ and the ASR?

What is the baseline ASR when no poison samples are injected?

---

### Official Review · Reviewer_rras · 2023-11-02

**Soundness:** 3 good
**Presentation:** 2 fair
**Contribution:** 2 fair
**Rating:** 3
**Confidence:** 4

**Summary:**

The authors propose a backdoor poisoning attack wherein poisons are crafted via an $l_\infty$ perturbation using a surrogate network, and triggers at inference time are crafted using this same pretrained network in order to cause some desired behavior in the victim model.

**Strengths:**

* The authors seem to do a thorough job of experimentation, testing on multiple architectures and including results against some defenses.
* For their threat model, the authors do seem to achieve the best when compared to the work they cite.

**Weaknesses:**

* I don't think the negative, neutral trigger options are a selling point for this paper, and any application I can think of these seems quite contrived.
* I'm also not a huge fan of the threat model wherein $\ell_\infty$ perturbations are added to the poisons, and again at inference time. I think patch attacks are a more convincing setting for backdoors.
* Along this same line, I'm not sure I like the positioning of this paper as a backdoor poisoning paper, as it's closer to a (boosted?) black box adversarial attack. If I understand correctly, the inference time perturbations are all different, and also separate from any perturbations added to poisons. So is the explanation of this method that injecting adversarial features during training makes adversarial attacks easier downstream?
* Many of the figures are way too small. Readers shouldn't have to significantly zoom in to view them. Move some to the appendix.
* It seems like you include a few results for $\ell_\infty$ bounds of $8/255$ in the appendix, but in the main body, it seems like you use a slightly looser $0.05$ bound. I would really stick to $8/255$ as this is the literature standard.
* Given the specific form of this attack's trigger, it would seem natural to experiment with adversarial training as a defense.
* It might be interesting to also compare to some hidden trigger backdoor attacks like [1,2] where instead of a patch trigger, you can adapt the method to a fixed $\ell_\infty$ trigger.

[1] Saha, Aniruddha, Akshayvarun Subramanya, and Hamed Pirsiavash. "Hidden trigger backdoor attacks." Proceedings of the AAAI conference on artificial intelligence. Vol. 34. No. 07. 2020.
[2] Souri, Hossein, et al. "Sleeper agent: Scalable hidden trigger backdoors for neural networks trained from scratch." Advances in Neural Information Processing Systems 35 (2022): 19165-19178.

**Questions:**

* I'm a bit confused by the trigger generator part of Fig 2. Why are the images of the green bird on the black background ($x_i$) perturbed to be a totally different bird ($\bar{x_i}$)? I thought the trigger generator also used an $\ell_\infty$ constraint at inference time?